# Behavior of Primary Human Oral Keratinocytes Grown on Invisalign® SmartTrack® Material

Michael Nemec [1], Hans Magnus Bartholomaeus [2], Christian Wehner [3], Christian Behm [1,2], Hassan Ali Shokoohi-Tabrizi [3], Xiaohui Rausch-Fan [2,3], Oleh Andrukhov [2,*] and Erwin Jonke [1]

[1] Division of Orthodontics, University Clinic of Dentistry, Medical University of Vienna, 1090 Vienna, Austria; michael.nemec@meduniwien.ac.at (M.N.); christian.behm@meduniwien.ac.at (C.B.); erwin.jonke@meduniwien.ac.at (E.J.)

[2] Competence Center for Periodontal Research, University Clinic of Dentistry, Medical University of Vienna, 1090 Vienna, Austria; hans.bartholomaeus@meduniwien.ac.at (H.M.B.); xiaohui.rausch-fan@meduniwien.ac.at (X.R.-F.)

[3] Division of Conservative Dentistry and Periodontology, University Clinic of Dentistry, Medical University of Vienna, 1090 Vienna, Austria; christian.wehner@meduniwien.ac.at (C.W.); hassan.shokoohi-tabrizi@meduniwien.ac.at (H.A.S.-T.)

\* Correspondence: oleh.andrukhov@meduniwien.ac.at; Tel.: +43-1-40070-2620

**Abstract:** Orthodontic clear aligner treatment is gaining tremendous popularity. The world market leader is Align Technology® and its product Invisalign®. Although numerous patients are treated with Invisalign® aligners, only little is known about the cellular effects of aligner material on oral epithelial cells. In the present study, we aimed to investigate the effects of SmartTrack® clear aligner material on directly cultured primary human oral keratinocytes (HOKs). Cell morphology and behavior were investigated by scanning electron microscopy and bright field microscopy. Aligner effects on viability were detected by cell-counting-kit (CCK)-8 and live/dead staining. Gene expression of several inflammatory and barrier proteins was assessed by qPCR. Cells cultured on tissue culture plastic served as control. Cell proliferation/viability was significantly lower in cells cultured on aligner material ($p < 0.05$) in comparison to control. Live/dead staining did not reveal an increase in the number of dead cells on aligner surfaces. After two and seven days of incubation, interleukin (IL)-6 expression decreased, and IL-8 expression increased in HOKs cultured on aligner surfaces. The expression of intercellular adhesion molecule 1 (ICAM-1) significantly decreased after seven days. Gene expression of epithelial barrier markers showed that integrin (ITG)-$\alpha$6 significantly decreased after two and seven days. A significant decrease in ITG-$\beta$4 and E-cadherin expression levels compared to control could only be seen after seven days. We did not find any cytotoxic effect, but alterations in the cell's barrier functions and inflammatory reaction were obvious. Clinical studies are required to give further insights into clinical reactions on the underlying aligner material of this quickly expanding orthodontic appliance.

**Keywords:** orthodontics; aligner; proliferation; epithelial barrier; inflammation; human oral keratinocytes; *in vitro*; Invisalign; SmartTrack

## 1. Introduction

Clear aligners, a series of clear plastic appliances, represent an esthetic and removable alternative to conventional fixed orthodontic treatment. Reinforced by the patients' request for less visible orthodontic devices and a systematic promotion policy of aligner companies, an increased demand for clear aligner treatment took place in the last decade [1–3]. This led to an intensified development and application of this orthodontic treatment modality [4]. Constant improvements in clear aligner technology resulted in increased numbers and complexity of aligner treatments [5,6]. Based on increasing ranges of aligner application, orthodontists and general dentists use aligner techniques to treat orthodontic patients [7,8].

Taken together, clear aligner therapy has moved in the clinical and scientific spotlight of the past years.

Align Technology is the most dominant aligner company in the orthodontic aligner market, and its product Invisalign® is with more than 9 million aligner treatments market leader (https://www.aligntech.com/about accessed on 20 March 2021). SmartTrack®, the latest Invisalign® material, consists of polyurethane and co-polyester. Invisalign® clear aligners are produced by a thermoforming process. Each aligner is produced based on a 3D printed dental cast, reflecting the actual clinical situation for the corresponding aligner [9]. SmartTrack® consists of a flat surface on the outside and a rough surface on the inside. Each aligner leads to a gradually improving clinical situation during treatment and should be worn for 22 h per day [10].

To act as effectively as possible, aligners should only be removed for oral hygiene and food intake and, thus, remain most of the time in direct contact with vital oral structures. Considering that an average orthodontic treatment takes multiple months, clear aligners could affect oral health [11]. Therefore, biocompatibility and material safety are pivotal aspects of long-lasting treatments [12]. To depict the effects of aligner material on oral cells, aligner eluates have been studied on multiple cells, including gingival fibroblasts, keratinocytes, and breast cancer cells. These studies reported a relatively weak cytotoxic effect of aligners under *in vitro* conditions [13–15]. Further, a recent meta-analysis reported inconsistency in the current literature concerning safety considerations. Further clinical and laboratory studies were needed to shed led on the biological impact of this emerging orthodontic field [3].

Oral health is mainly depending on the integrity of the epithelium, a barrier of the body to its environment. In general, the oral epithelium is comprised of two main structures: the surface stratified squamous epithelium and its deeper basement membrane with the underlying connective tissue including nerves, lymphatic and blood vessels [16]. Gingiva is a tooth surrounding tissue that is set up by connective and epithelial tissues. Its epithelium is stratified squamous keratinized in the region of the gingiva and the papillae [17]. As outermost cell type in the keratinized part of the oral cavity, oral keratinocytes play a pivotal role in the oral defense system [18].

Since clear aligners are directly in contact with oral epithelial structures and no data existed on the behavior of oral cells directly grown on solid clear aligner specimens, we described in a previous *in vitro* study the effects of SmartTrack® material on human oral epithelial cells directly grown on the material [19]. This study was performed with oral squamous carcinoma (Ca9-22) cells to give a first insight into the SmartTracks®' cellular effects under direct contact conditions. We found no evidence of negative effects on oral squamous carcinoma cells. Although oral squamous carcinoma cells are often used as a model of the oral epithelium in *in vitro* research, these cells do not entirely reflect all properties of oral epithelium. There are some critical differences between oral squamous carcinoma cells and normal oral keratinocytes in terms of proliferation, inflammatory response, and gene expression [20,21]. Therefore, the biocompatibility of Invisalign® aligners needs to be also confirmed in the experiment with non-tumor cells.

Thus, in the present study, we aimed to investigate the effects of clear aligner material on directly cultured primary human oral keratinocytes. We focused on cell attachment, proliferation/viability, cell death, and gene expression of several functional proteins involved in the epithelial defense function. Since epithelial defense is realized by means of the inflammatory response and mechanical barrier, we have focused on the inflammatory parameters interleukin (IL)-1β, tumor necrosis factor (TNF)-α, IL-6, IL-8 and intercellular adhesion molecule (ICAM)-1, as well as proteins involved in barrier function integrin (ITG)-α6, ITG-β4, and E-cadherin.

## 2. Materials and Methods

### 2.1. Aligner Preparation and Cell Culture Stimulation Protocol

Aligner discs were prepared and characterized as previously described [19].



Pooled primary human oral keratinocytes (HOKs) were purchased from CELLnTEC (Bern, Switzerland) and cultivated as recommended by the manufacturer. Briefly, HOKs were cultured in keratinocytes medium CnT-PR (CELLnTEC, Bern, Switzerland) at 37° Celsius, 5% $CO_2$, and 95% humidity. Aligner discs were put into the wells of 96-well plates and were fixed using colorless high-vacuum silicone grease (Sigma-Aldrich, St. Louis, MO, USA). $1 \times 10^4$ HOKs, re-suspended in 15 μL of CnT-PR medium, were seeded on both discs' surfaces. After four hours' incubation additional 85 μL of CnT-PR medium was added to each well. Two and seven days later, changes in cell morphology were investigated by scanning electron microscopy. A potential effect of aligners on cell behavior and viability was evaluated by the CCK-8-based cell viability assay and live/dead staining. The effects of aligner's discs on the expression of inflammatory and cell adhesion parameters were investigated by quantitative polymerase chain reaction (qPCR). For all experiments, HOKs seeded on tissue culture plastic (TCP) served as a negative control.

### 2.2. Scanning Electron Microscopy

After two and seven days of incubation, $1 \times 10^4$ HOKs on both aligners' surfaces were used to analyze the cells' morphology and microstructure by scanning electron microscopy (SEM, Quanta 200, FEI, Hillsboro, OR, USA). HOKs were fixed with 4% formaldehyde for 24 h, followed by washing the cells with phosphate-buffered saline (PBS) three times. Subsequently, dehydration was performed by rinsing HOKs with ethanol (Merck, Darmstadt, Germany), using gradually increased ethanol concentrations. The last rinse was performed with hexamethyldisilazane (Sigma-Aldrich, St. Louis, USA), which was followed by coating the cells with a sputter coater (EM ACE200, Leica, Wetzlar, Germany), generating a gold layer with 100 nm. The cells were observed under the SEM acquiring surface and cross-sectional views and using a 400- and 1500-fold magnification and an accelerating voltage of 15 kV. For each preparation type, triplicates were analyzed.

### 2.3. Bright Field Microscopy

Microscopic analysis was performed two and seven days after the cultivation of HOKs on aligner discs as described above. Cells were detected on TCP, inner and outer aligner material. Images were taken by a bright-field microscope (Revolve, Discover Echo Inc., San Diego, CA, USA) under fourfold magnification.

### 2.4. Cell Proliferation/Viability

Cell proliferation/viability was evaluated by using the cell counting kit 8 (CCK8) photometric assay after two and seven days of incubation. After cultivating HOKs as described above, 10 μL of CCK-8 reagent were added per well, followed by incubation for 4 h at 37° Celsius. Afterward, 80 μL of conditioned media were transferred to a new 96-well plate, and the optical density at 450 nm (OD450 nm) was measured using the Synergy HTX photometer (Bioteck, Winooski, VT, USA). Four independent cell viability experiments were performed in triplicates.

### 2.5. Live/Dead Staining

Two and seven days after culturing $1 \times 10^4$ HOKs on aligners' disks, cell viability was evaluated by using the Live/Dead Cell Staining Kit (Enzo Life Sciences, Framingdale, NY, USA), according to the manual. Briefly, adherent HOKs were washed with $1 \times$ PBS (phosphate-buffered saline) three times, followed by staining the cells with 100 μL/well staining solution (1 μL solution A + 1 μL solution B in 1 mL staining buffer). After 15 min of incubation at 37° Celsius, fluorescent cells were detected by using a fluorescent microscope (Nikon, Tokyo, Japan). A four-fold magnification and an exposure time of 80 ms were used to determine green fluorescent Live-Dye™ (Ex/Em = 488/518 nm) and propidium iodide (PI, Ex/Em = 488/515 nm). The live/dead staining was performed in triplicates.

*2.6. Quantitative Polymerase Chain Reaction (qPCR)*

Gene expression analysis was made in HOKs cultured similarly to the protocol mentioned above for two and seven days. HOKs cultured on TCP served as the negative control. TaMan® Gene Expression Cells-to-Ct kit (Ambion/Applied Biosystems, Foster City, CA, USA) was used for cell lysate preparation, mRNA transcription into cDNA, and qPCR. The reverse transcription of mRNA into cDNA was executed with the following settings: 37° Celsius for one hour and 95° Celsius for 5 min, followed by 4° Celsius using the Primus 96 advanced thermocycler (PeqLab/VWR, Darmstadt, Germany). qPCR was performed on an ABI SepOnePlus device (Applied Biosystems, Foster City, CA, USA) using the following thermocycler settings: $1 \times 95°$ Celsius for 10 min followed by 50 cycles 15 s at 95° Celsius and 60° Celsius for one minute per cycle. The following TaqMan gene expression assays were used (all from Applied Biosystems, Foster City, CA, USA): IL-1β, Hs01555410_m1; TNF-α, Hs99999043_m1; IL-6, Hs00985639_m1; IL-8, Hs00174103_m1; ICAM-1, Hs00164932_m1; ITG-α6, Hs01041011_m1; ITG-β4, Hs00173995_m1; E-cadherin, Hs01023894_m1; glyceraldehyde 3-phosphate dehydrogenase (GAPDH), Hs99999905_m1. Cycle threshold (Ct) values were determined for each gene. The expression of target genes was calculated as n-fold expression compared to negative control using GAPDH as the endogenous reference by $2^{-\Delta\Delta Ct}$ method. All qPCR reactions were performed at least in triplicates.

*2.7. Statistical Analysis*

Statistical analysis was performed using SPSS 23.0 (SPSS Inc., Chicago, IL, USA). All data are presented as mean $\pm$ standard error of the mean (S.E.M.) from four independent experiments. The Kolmogorov-Smirnov test was used to verify the normal distribution of all received data. Statistical differences were determined by one-way analysis of variance (ANOVA) for repeated measures using post-hoc LSD test for pairwise comparisons. Differences with *p*-values less than 0.05 were considered to be statically significant.

## 3. Results

*3.1. Microscopic Analysis*

Figure 1 shows representative SEM images after two days of culture, taken with two different magnifications. All investigated surfaces show similar cell distribution at both magnifications. Representative bright field microscopy images of HOKs cultured on different surfaces are presented in Figure 2. Cells cultured on the inner aligner surface reveal a smaller amount of cells compared to the outer surface. This effect can be seen after two and seven days of incubation. In general, HOKs show a lower cell density on aligner surfaces compared to TCP at both time points.

*3.2. Proliferation/Viability of HOKs*

Figure 3 shows the proliferation/viability of HOKs grown on different aligner surfaces for two and seven days. HOKs grown on both aligner surfaces show significantly lower proliferation/viability than compared to those grown on TCP ($p < 0.05$). However, no statistically significant difference between both aligner surfaces was observed after both two and seven days.

*3.3. Live/Dead Staining*

Live/dead staining images are presented in Figure 4. Representative images display HOKs grown on TCP, inner and outer aligner surface after two and seven days of culture. Most cells remain viable during the observation time. There was only a relatively low amount of dead cells. Cells cultured on the inner aligner surface showed a lower cell number than the outer aligner surface or TCP. HOKs distributed evenly, and only some cluster-like structures were observed.

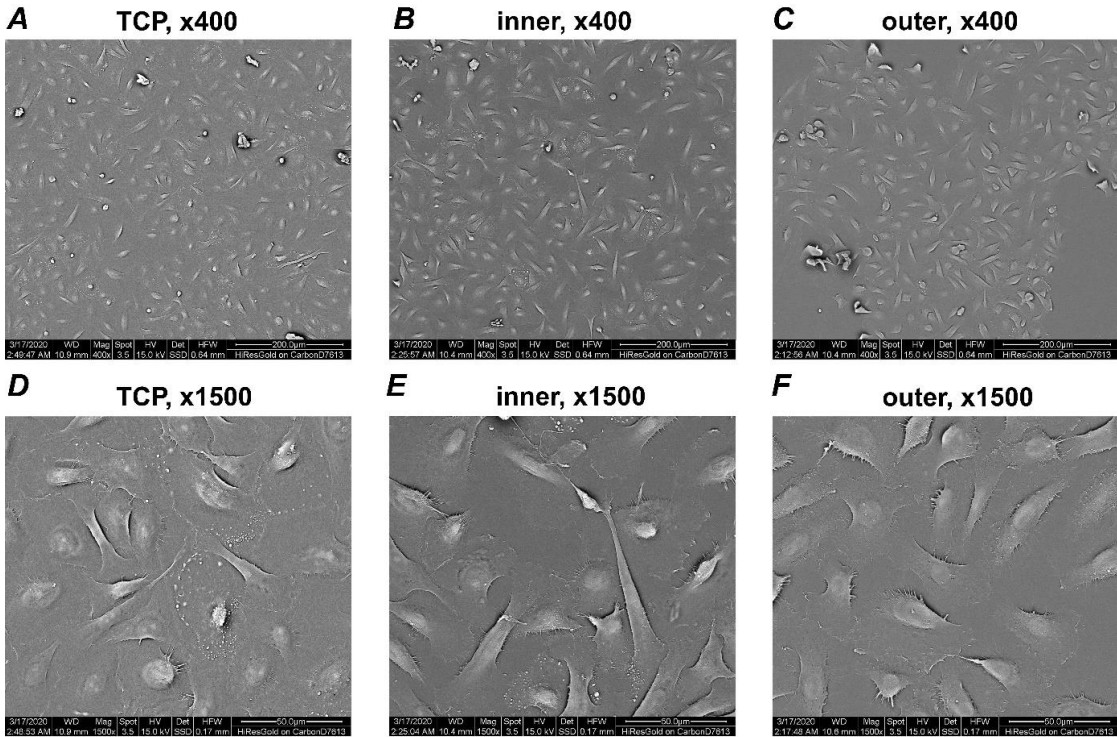

**Figure 1.** Scanning electron microscopy analysis of HOKs. HOKs were grown on two different aligner surfaces. Pictures were taken at two different magnifications (upper row, $400\times$; lower row, $1500\times$). Cells were seeded on TCP as control (**A**,**D**), inner (**B**,**E**) and outer (**C**,**F**) aligner surfaces. Pictures were taken after 2 days of culture. Scale bars correspond to 200 μm (**A**–**C**) or 50 μm (**D**–**F**).

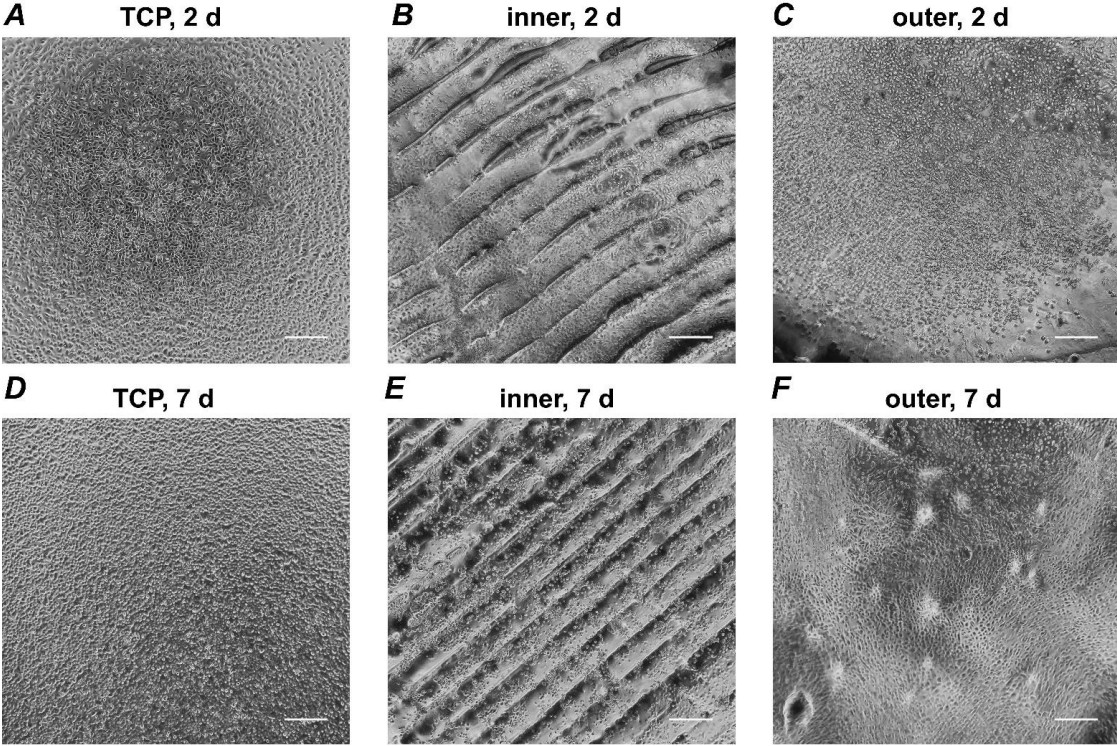

**Figure 2.** Brightfield microscopy images of HOKs. HOKs were grown on two different aligner surfaces and TCP as control. Pictures were taken at four-fold magnification. Cells were seeded on TCP as control (**A**,**D**), inner (**B**,**E**) and outer (**C**,**F**) aligner surfaces. Pictures were taken after 2 (upper row) and 7 days of culture (lower row). Scale bars correspond to 200 μm.

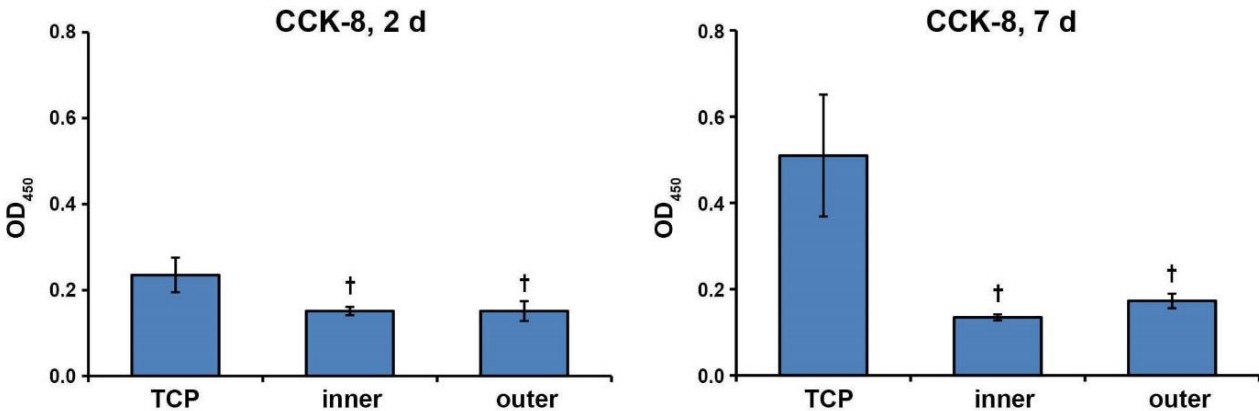

**Figure 3.** Proliferation/viability of HOKs grown on different aligner surfaces. HOKs were cultured on TCP, inner and outer aligners' surfaces. Proliferation/viability was measured after 2 and 7 days of incubation by CCK-8 assay. Cells grown on tissue culture plastic served as control. *Y*-axis represents the optical density (OD) measured at 450 nm. Data represents mean ± S.E.M. of three independent experiments. †—significantly lower compared to control; $p < 0.05$.

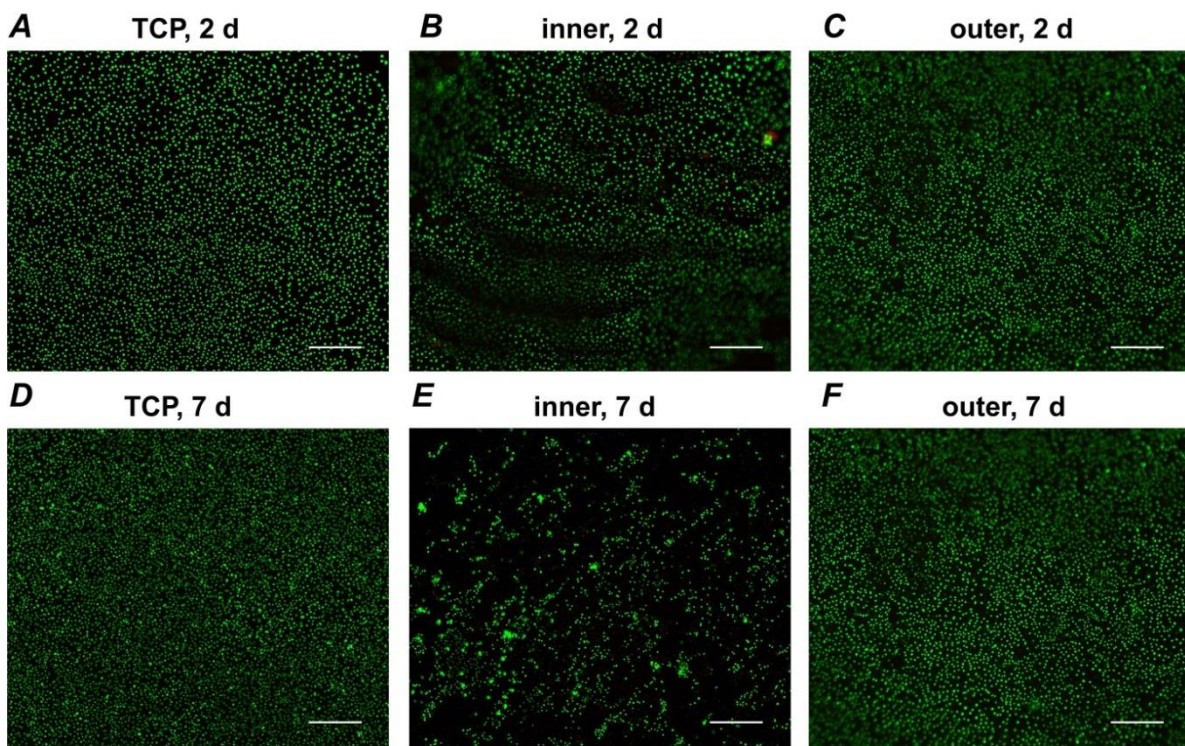

**Figure 4.** Live/dead Staining. HOKs were grown on TCP (**A**,**D**), inner (**B**,**E**), and outer (**C**,**F**) aligner surfaces for 2 and 7 days and were stained with Live/dead staining kit. Vital cells are visible as green, while the dead cells are presented red. Images are taken from a representative experiment. Scale bar corresponds to 200 µm.

### 3.4. Gene Expressions of Inflammatory Markers

The gene expression levels of IL-1β, TNF-α, IL-6, IL-8, and ICAM-1 in HOKs grown on the different surfaces for two and seven days are shown in Figure 5. After two and seven days of incubation, IL-1β significantly increased if cultured on the outer aligner surface, whereas IL-6 significantly decreased if cultured on both aligner surfaces. However, there were no differences in the gene expression level of IL-6 regarding the aligner surface. Regarding TNF-α, HOKs showed significantly decreased gene expression levels after two and seven days.

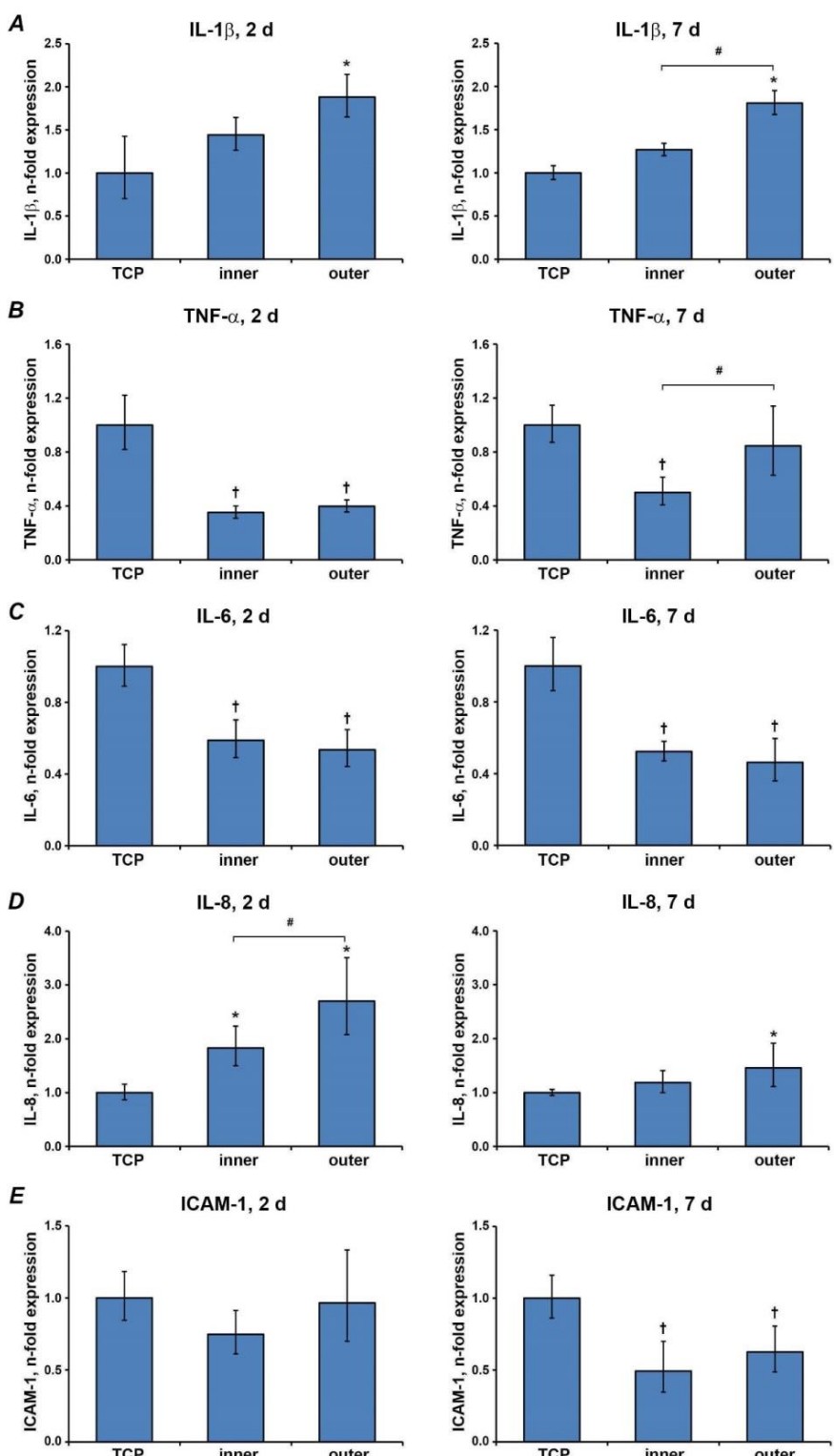

**Figure 5.** Gene expression of inflammatory markers in HOKs grown on different surfaces. Gene expression of IL-1β (**A**), TNF-α (**B**), IL-6 (**C**), IL-8 (**D**) and ICAM-1 (**E**) after 2 and 7 days in HOKs grown on different aligners' surfaces and TCP were measured by qPCR. Y-axes represent n-fold expression in relation to HOKs grown on TCP (n-fold expression = 1), which was calculated using the $2^{-\Delta\Delta Ct}$ method. Data are presented as the mean ± S.E.M. of four independent experiments. *—significantly higher compared to control; †—significantly lower compared to control; #—significantly different between groups; $p < 0.05$.

After two days, significant increases in IL-8 expression levels could be detected for both aligner surfaces. Whereas, after seven days, only the outer surface showed a statistically significant increase in gene expression levels in comparison to control. Differences between the two different aligner surfaces regarding the gene expression level of IL-8 could be detected after two days with a higher expression in cells cultured on the outer aligner surface. ICAM-1 gene expression levels showed only after seven days a statistically significant decrease on both aligner surfaces in comparison to control.

### 3.5. Gene Expression Levels of Proteins Involved in the Barrier Function

The effect of different aligner surfaces on the expression of ITG-α6, ITG-β4, and E-cadherin is shown in Figure 6. Gene expression levels of ITGα-6 significantly decreased after two and increased after seven days for both aligner surfaces. A significant decrease in ITG-β4 expression levels could only be seen after seven days for the outer aligner surface. E-cadherin showed a similar effect compared to ITG-β4 ($p > 0.05$). All parameters showed no statistically significant differences between different aligners surfaces ($p > 0.05$).

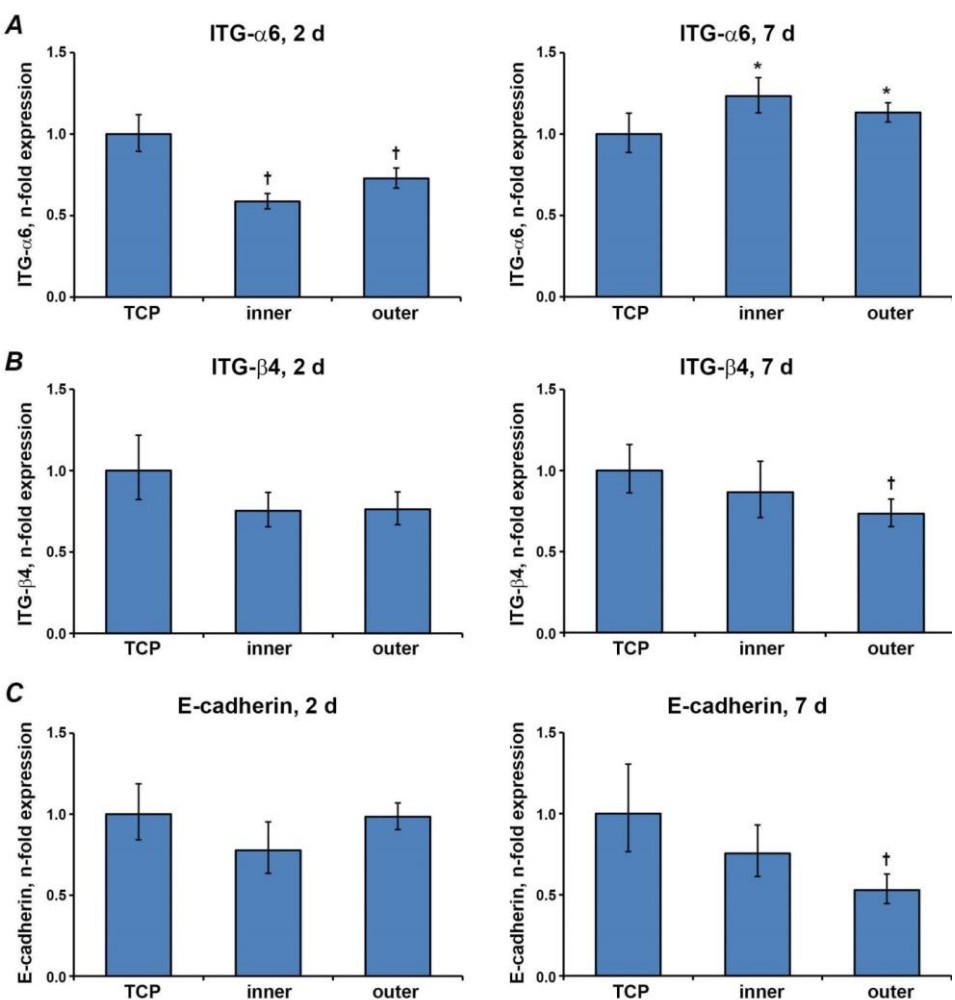

**Figure 6.** Gene expression of epithelial barrier markers in HOKs grown on different surfaces. Gene expression of ITG-α6 (**A**), ITG-β4 (**B**) and E-cadherin (**C**) after 2 and 7 days in HOKs grown on different aligners' surfaces and TCP were measured by qPCR. Y-axes represent n-fold expression in relation to HOKs grown on TCP (n-fold expression = 1), which was calculated using the $2^{-\Delta\Delta Ct}$ method. Data are presented as the mean ± S.E.M. of four independent experiments. *—significantly higher compared to control; †—significantly lower compared to control; $p < 0.05$.

## 4. Discussion

Most previous studies of aligner effects on oral cells used eluate settings [13–15]. In our recent study, we investigated for the first time the behavior of oral squamous carcinoma cells directly grown on SmartTrack® aligner material and found no adverse effect on these cells [19]. Oral squamous carcinoma cells represent a well-established *in vitro* model for test settings of oral epithelial cells [22–24] but have their limitations compared to primary oral epithelial cells. Thus, oral squamous carcinoma cells are less susceptible to apoptosis than primary epithelial cells [25], and their expression of some functional properties, particularly E-cadherin, is altered during tumor transformation [26]. Therefore, it has to be proven if the investigated effects of Invisalign® clear aligner material can be also be seen in non-neoplastic oral epithelial cells. In the gingival papilla region, a tissue designed for peripheral body defense [27], clear orthodontic aligners are in close contact with gingival epithelium. Oral keratinocytes are the main cell type in the gingival epithelial tissue, a barrier that separates the body from its environment Thus, we investigated the effect of SmartTrack® aligner material on HOKs in this study and tested for proliferation/viability, morphology, and the expression of various proteins involved in the epithelial barrier function and local inflammatory response.

An important observation of our study is that primary epithelial cells, similarly to oral squamous carcinoma cells [19], do not grow on aligner surfaces. This was proven by assessing the proliferation/viability of epithelial cells using the CCK-8 method. Similar to Ca9-22, no toxic effect of aligner surfaces on cell viability was observed in HOKs. Invisalign® surfaces and tissue culture plastic exhibited similar hydrophilicity as measured by contact angles [19]. Therefore, the fact that HOKs grew on Invisalign® surfaces exhibiting low proliferation and different attachment and behavior compared to TCP could be related mainly to the different compositions of these materials. It is reported that the growth of different cell types, including keratinocytes, decreases for polyurethane compared to TCP [28]. Our findings regarding proliferation/viability are in accordance with Premaraj et al., who demonstrated decreased metabolic activity in immortalized human keratinocyte N/TERT-1 cell line using Invisalign® eluates in saline solution and artificial saliva [14].

Microscopic analysis after two and seven days showed no substantial differences after both time points. This can be seen in both bright field microscopy and scanning electron microscopy images. Furthermore, we observed that HOKs distributed evenly, and some cluster-like structures were observed rather rarely. In contrast, in our previous study, Ca9-22 cells tend to cluster formation, especially after seven days of culture [19]. This clustering effect could rather be explained accordingly by an intrinsic cell reaction than by a material-dependent cell behavior.

Oral epithelium acts as a mechanical barrier for the infection and is involved in the immune response [29]. Therefore, we focused on the gene expression of some proteins involved in these two crucial functions. Live/dead staining did not reveal any substantial changes in the overall amount of dead cells on aligner material. It should be noted that PI, which is used in live/dead staining, recognizes only the cells in the late apoptosis. Additional staining with annexin V might be necessary to distinguish early apoptotic cells. However, because the number of PI$^+$ cells was very low, we do not expect any influence of apoptosis on gene expression results.

Oral keratinocytes express a variety of cytokines and chemokines, such as IL-1$\alpha$, IL-1$\beta$, TNF-$\alpha$, IL-6, and IL-8 [30–32]. Besides, these inflammatory mediators are also involved in the development of gingivitis [29], a commonly reported side effect of orthodontic aligner therapy [33,34]. The major finding of our study is that Invisalign® aligners alter the balance between different inflammatory mediators.

We found a decrease in TNF-$\alpha$ and IL-6 expression in HOKs grown on the aligners. A similar trend could also be seen for ICAM-1. TNF-$\alpha$ is produced by epithelial cells in response to infection, especially at the initial infection phase [35]. Dysregulation of IL-6 is associated with numerous oral diseases, including periodontal disease, oral cancer and lichen planus [36]. ICAM-1 is responsible for leukocyte adhesion and their transepithelial

migration [37]. In contrast, the expression of IL-1β and IL-8 increased after culturing HOKs on the aligners. Similarly to TNFa, IL-1β is also produced by oral epithelium upon exposure to the pathogens and plays an important role in promoting the inflammatory response. IL-8 is an important chemoattractant, and even a small amount of IL-8 stimulates leukocyte migration [37,38]. Dysregulation of leukocyte migration through IL-8 degradation by periodontal pathogen *Porphyromonas gingivalis* is an important factor of periodontal disease pathogenesis [39].

Translation of our data to the clinical situation is rather tricky because both pro- and anti-inflammatory effects of aligners on HOKs were observed. For example, an increased expression of IL-8 might enhance infiltration of leukocytes, promote the inflammatory reaction, and therefore represents a risk factor for oral health [40,41]. Similarly, enhanced production of IL-1β could be considered as a potential risk factor for the aligner therapy. To some degree, these pro-inflammatory effects might be balanced by the anti-inflammatory action of aligners. However, taken together, alterations in the balance between different inflammatory mediators will impact the host-microbe homeostasis. In the case of accompanying inflammatory diseases such as gingivitis, an altered inflammatory response could lead to faster progress in disease development.

Our experiments with HOKs show that inflammatory parameters are not affected as much as it was concluded from our previous study using Ca9-22 cells [19]. Our previous study found substantial increases of all inflammatory parameters under the same experimental conditions [19], indicating different molecular responses by different types of epithelial cells to SmartTrack® aligner material. Furthermore, we have mentioned that HOKs, grown on the inner surface, generally exhibit lower levels of pro-inflammatory mediators compared to those grown on the outer surface. This finding might also have some relevance because in the clinical situation, only the inner surface has contact with oral epithelium.

The barrier function of oral epithelium is realized through the interconnections of keratinocytes by various transmembrane proteins [42]. This mechanical barrier is the first layer of defense against exogenous noxious agents and bacterial invasion [43]. Among transmembrane proteins, integrins play a pivotal role in keratinocytes for cell adhesion to the extracellular matrix and cell detachment [44]. There is a strong relation between trans-mucosal resistance and the number of intercellular junctions [29]. E-Cadherin, a transmembrane protein, plays an important role in the maturation and formation of these intercellular junctions. Hence, to assess the potential effect of aligners' material on the cell barrier function, we have investigated the gene expression of E-cadherin, ITG-α6, and ITG-β4. After two days of incubation, no significant effect on epithelial barrier function parameters could be detected compared to TCP. After seven days, only the expression of ITG-α6 increased, whereas E-cadherin and ITG-β4 expression significantly decreased. This expression pattern in HOKs differs strongly from oral squamous carcinoma cells. Ca9-22 cells grown on aligners exhibited significantly higher gene expression levels of E-cadherin, ITG-α6, and ITG-β4 after two and seven days with a more pronounced effect after two days compared to seven days [19]. Regarding the expression of parameters associated with barrier function in HOKs, we concluded that SmartTrack® aligner material might impair epithelial barrier function in HOKs. In oral squamous carcinoma cells, we did not find any inhibitory effects by the underlying aligner material and, therefore, no apparent risk for barrier function [19]. However, a potentially harmful effect of the Invisalign® appliance on oral epithelium barrier function should be proved by clinical studies.

## 5. Limitations

This study aimed to depict the effects on Invisalign®SmartTrack®material in human oral keratinocytes in vitro. However, this does not necessarily explain the situation in vivo. Further, commercially available cells were cultured in corresponding media, recommended by the manufacturer, whereas in the oral cavity, the contact between aligners and oral epithelium takes place in the presence of saliva and various other exogenous factors, such

as bacteria and mineral deposits. Thus, it would be interesting to investigate the primary epithelial cells from the patients undergoing orthodontic treatment with aligners to give more insights into their effects during clinical use. Furthermore, epithelial cells do not grow on aligner material under in vivo conditions. In the clinical situation, aligners contact with already formed epithelial layer. We have chosen the model with direct cell growth on aligner material because it was the best way to bring cells in direct contact with Invisalign material. Compared to TCP, obtained aligner discs were cut out from aligner material and cannot exhibit similar flatness. Therefore, a cell monolayer could tend to have more pronounced contact to the underlying material in the most convex region of the aligner disc. This situation may best reflect the clinical region of close contact between the gingival papilla and aligner material, where the aligner can soundly cover the outermost cell layer.

## 6. Conclusions

The current literature does not give enough insights into clear aligner effects on oral cells so far [3]. Our previous study and the underlying results on HOKs show that there could be some risks for oral health, which can be argued by the contact to the gingival papilla, as Invisalign®aligner material is in close contact with the gingival papilla. Therefore, aligner material could interfere with the function of oral epithelium and should be carefully investigated in further clinical studies. Supposed HOKs can be seen as a closer model to clinical conditions than oral squamous carcinoma cells. In that case, the results of this study could be carefully used for establishing a future clinical setting. It has to be proven if the results of the underlying study add additional information on potential clinical effects.

**Author Contributions:** Conceptualization, M.N., X.R.-F. and O.A.; methodology, M.N., H.M.B., C.B. and O.A.; validation, M.N. and O.A.; formal analysis, M.N., H.M.B., O.A. and X.R.-F.; investigation, M.N., H.M.B., C.W. and H.A.S.-T.; resources, E.J. and X.R.-F.; data curation, M.N.; writing—original draft preparation, M.N. and O.A.; writing—review and editing, H.M.B., C.W., C.B., H.A.S.-T., E.J., X.R.-F.; visualization, H.A.S.-T. and O.A.; supervision, E.J., O.A. and X.R.-F.; project administration, X.R.-F.; funding acquisition, M.N. and O.A. All authors have read and agreed to the published version of the manuscript.

**Funding:** The research was supported by Medical-scientific fund of mayor of Vienna ("Medizinisch-Wissenschaftlicher Fonds des Bürgermeisters der Bundeshauptstadt Wien", project 19089).

**Institutional Review Board Statement:** Not applicable because commercially available cell line was used.

**Informed Consent Statement:** Not applicable.

**Data Availability Statement:** The data presented in this study are available on request from the corresponding author.

**Acknowledgments:** The authors would like to thank Phuong Quynh Nguyen and Hedwig Rutschek for their support throughout the study.

**Conflicts of Interest:** All authors declare no conflict of interest.

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
