# Peer review of "Behavior of Primary Human Oral Keratinocytes Grown on Invisalign® SmartTrack® Material"

_applsci, doi:10.3390/app11062826_

Round 1
Reviewer 1 Report
- Content beginning in line 98 and line 104 can be combined. Mentions of methods already addressed in detail later in the section should be removed.
- Figure 1 and 2 show the same results and should be combined as one figure.
- Figure 3 should be present as %cell viability or similar
- Figure 4 offers qualitative data but quantification and results of quantification should be included to in order to support the results.
- Other experiments to determine changes in cell viability should be included.
- for example annexin staining or protein expression of early markers for cell death.
- Justification and background on why the genes shown were chosen.
- For example oral keratinocytes produce a large number of inflammatory factors such as IL-1 and TNF that were not included in the panel why were IL-6 and IL-8 chosen. same with ITG-a6 and ITG-b4
- The experimental set up of growing the cells directly on the material is not a relevant model for in vivo barrier function since in a clinical setting the cells are not growing on the material. Cell barrier should be formed first and then exposed to material in order to study effects on barrier function.
- The references are not numbered correctly. Line 311 has citation number 21 listed but the reference does not correspond to the content.
- the layout of the references needs improvement.
- Throughout the discussion there is no attempt to explain the results and the implications of the results in depth.
- For example in 298-300 it mentions that there was an effect with IL-6 expression but not IL-8, there is no discussion addressing why one would be upregulated but not the other and what implications elevated IL-8 has.
- Several formatting issues
- Spacing needs to be consistent throught
- For example line 272 is a space but the rest of the paragraphs in that section do not have the same spacing.
- Line 201: Change Figure X
- In the methods line 136 is it 14hrs or 1-4hours
- Spacing needs to be consistent throught
Author Response
We are thankful to Reviewer 1 for critical evaluation of our manuscript and constructive criticism. Please, find below our point-by-point answer to all raised issues.
- Content beginning in line 98 and line 104 can be combined. Mentions of methods already addressed in detail later in the section should be removed.
The content of these two paragraphs has been combined.
- Figure 1 and 2 show the same results and should be combined as one figure.
Thank you for this comment. In general, we agree that these two Figures have somewhat similar output. However, there are also certain differences between them. Figure 1 shows cells after 2 days of culture visualized with SEM at two different magnifications and aims to demonstrate that cells have real contact with the Invisalign surface. In contrast, Figure 2 shows the cells at two different time points taken under the usual bright-field microscope. These images were taken at lower magnification and are aimed to demonstrate the overall picture of cells growing on different surfaces. For these reasons, we would like to keep these Figures separate and hope that our thinking would also be acceptable for Reviewer.
- Figure 3 should be present as %cell viability or similar
Thank you very much for stressing this point. We have considered the possibility to present cell viability in percentages, but in our view, such presentation would have certain limitations. First, in Figure 3, one can clearly see that HOKs are growing only on TCP but not on aligner by comparing the data from 2 and 7 days. This information would be lost when the data are presented as the percentage of TCP. Second, this Figure shows the original data as measured by a photometer without any mathematical transformation, and this is also an advantage in the sense of objective presentation.
- Figure 4 offers qualitative data but quantification and results of quantification should be included to in order to support the results.
Thank you very much for this comment. To distinguish between living and dead HOKs, we used the Live-Dead Cell Staining Kit from Enzo followed by fluorescence microscopy as a qualitative approach. The aim of this experiment was to show that our experimental conditions have no apoptotic effect on HOKs. We could prove it because most of the cells were viable for up to seven days of culture. Unfortunately, living and dead cells cannot be counted with our microscope-based software to get a quantitative output.
- Other experiments to determine changes in cell viability should be included for example annexin staining or protein expression of early markers for cell death
Thank you very much for your criticism. We agree that the analysis of annexin V expression by either flow cytometry or microscopy and the analysis of specific protein expression in HOKs might provide some additional information regarding cell apoptosis. However, most of these methods are difficult to apply in our settings because we use a rather low cell number (only 10,000 cells per one disc), which makes it difficult to isolate a sufficient amount of protein or analyze it by flow cytometry. We have discussed the possibility of Annexin V staining in the Discussion section of our revised version by:
“It should be noted that PI, which is used in live/dead staining, recognizes only the cells in the late apoptosis. Additional staining with annexin V might be necessary to distinguish early apoptotic cells. However, because the number of PI+ cells was very low, we do not expect any influence of apoptosis on gene expression results.”
- Justification and background on why the genes shown were chosen.
Thank you for stressing this point. Since the epithelium functions as a mechanical barrier and causes an inflammatory response, we have focused in our study on proteins involved in these two functions. This fact is mentioned in the revised version of the manuscript (see, lines 101-106) by the following
"Since epithelial defense is realized by means of the inflammatory response and mechanical barrier, we have focused on the inflammatory parameters interleukin (IL)-1b, tumor necrosis factor (TNF)-α, IL-6, IL-8 and intercellular adhesion molecule (ICAM)-1, as well as proteins involved in barrier function integrin (ITG)-α6, ITG-β4, and E-cadherin."
Additionally, the functional importance of all these factors is discussed in Discussion section.
- For example, oral keratinocytes produce a large number of inflammatory factors such as IL-1 and TNF that were not included in the panel why were IL-6 and IL-8 chosen. same with ITG-a6 and ITG-b4.
Thank you for this constructive input. We performed further qPCR analysis of IL-1β and TNF-α expression. The data were included in the revised version of the manuscript (see, revised Figure 5.) and added the following:
"The gene expression levels of IL-1β, TNF-α, IL-6, IL-8, and ICAM-1 in HOKs grown on the different surfaces for 2 and 7 days are shown in Figure 5. After 2 and 7 days of incubation, IL-1β significantly increased if cultured on the outer aligner surface, whereas IL-6 significantly decreased if cultured on both aligner surfaces. However, there were no differences in the gene expression level of IL- 6 regarding the aligner surface. Regarding TNF-α, HOKs showed a significantly decreased gene expression levels after 2 and 7 days."
- The experimental set up of growing the cells directly on the material is not a relevant model for in vivo barrier function since in a clinical setting the cells are not growing on the material. Cell barrier should be formed first and then exposed to material in order to study effects on barrier function.
We agree with this critic point that our experimental model does not entirely reflect the clinical situation. This is definitively a limitation of the study. We added this point to the limitation section as follows:
"Furthermore, epithelial cells do not grow on aligner material under in vivo conditions. In the clinical situation, aligners contact with already formed epithelial layer. We have chosen the model with direct cell growth on aligner because it was the best way to bring cells in direct contact with Invisalign material. Compared to TCP, obtained aligner discs were cut out from aligner material and cannot exhibit similar flatness. Therefore, a cell monolayer could tend to have more pronounced contact to the underlying material in the most convex region of the aligner disc. This situation may best reflect the clinical region of close contact between the gingival papilla and aligner material, where the aligner can soundly cover the outermost cell layer."
- The references are not numbered correctly. Line 311 has citation number 21 listed but the reference does not correspond to the content.
Thank you for this comment. We adapted all references.
- the layout of the references needs improvement.
We improved the layout of all references.
- Throughout the discussion there is no attempt to explain the results and the implications of the results in depth. For example in 298-300 it mentions that there was an effect with IL-6 expression but not IL-8, there is no discussion addressing why one would be upregulated but not the other and what implications elevated IL-8 has.
Thank you for this comment. We adapted and extended the discussion in general and the role of inflammatory cytokines in particular. New and revised parts in discussion are highlighted in yellow.
- Several formatting issues
We took care of your input.
- Spacing needs to be consistent through
We took care of this as well.
- For example line 272 is a space but the rest of the paragraphs in that section do not have the same spacing.
We changed the spacing according to your recommendation.
- Line 201: Change Figure X
We adapted the number of the figure.
- In the methods line 136 is it 14hrs or 1-4hours
We changed it to 4 hours.
Reviewer 2 Report
Article is original and based on investigation of normal oral keratinocytes reactions to Invisalign® clear aligner. In my opinion such choose of cell line is more realistic than to use oral carcinoma cells. In the future it would be interesting to test effects of the orthodontic materials on the the other types of cells present in the submucosa including mesenchymal stem cells.
Authors in the investigation used substantial variety of contemporary methods: brightfield, scanning electron, fluorescent microscopy, and molecular markers.
Despite general good impression about the article I noticed several places where additional explanations or changes can improve the quality:
- Fig. 2. Very different pattern photographed after 2 and 7 days in inner and outer surfaces of the aligner material. This should be commented
- Fig.4 E no comments in the text about possible cell clusters formation. In my opinion cell clusters are visible in this picture
- Sentence located in the lines 86 and 87 should be edited.
Author Response
We are thankful to Reviewer 2 for overall positive evaluation of our work and criticism. Here are our answers to all raised points.
- 2. Very different pattern photographed after 2 and 7 days in inner and outer surfaces of the aligner material. This should be commented
Thank you for this constructive note. We have changed the photograph to a more representative one of the same time point.
- 4 E no comments in the text about possible cell clusters formation. In my opinion cell clusters are visible in this picture
Thank you for this input. We added the following to the results/discussion section (page. 8, line 224/225):
HOKs distributed evenly and only some cluster-like structures were observed.
- Sentence located in the lines 86 and 87 should be edited.
Thank you. We revised the sentence in lines 86 and 87.
Round 2
Reviewer 1 Report
No other corrections required.